# Development, deployment and evaluation of digitally enabled, remote, supported rehabilitation for people with long COVID-19 (Living With COVID-19 Recovery): protocol for a mixed-methods study

Elizabeth Murray ![ORCID],[1] Henry Goodfellow ![ORCID],[2] Julia Bindman,[1] Ann Blandford ![ORCID],[3] Katherine Bradbury,[4] Tahreem Chaudhry,[2] Delmiro Fernandez-Reyes ![ORCID],[5] Manuel Gomes,[6] Fiona L Hamilton,[1] Melissa Heightman,[7] William Henley,[8] John R Hurst ![ORCID],[9] Hannah Hylton,[10] Stuart Linke,[11] Paul Pfeffer,[10] William Ricketts ![ORCID],[10] Chris Robson,[12] Richa Singh,[10] Fiona A Stevenson,[1] Sarah Walker,[8] Jonathan Waywell[12]

**Correspondence to**
Professor Elizabeth Murray;
elizabeth.murray@ucl.ac.uk

## ABSTRACT

**Introduction** Long COVID-19 is a distressing, disabling and heterogeneous syndrome often causing severe functional impairment. Predominant symptoms include fatigue, cognitive impairment ('brain fog'), breathlessness and anxiety or depression. These symptoms are amenable to rehabilitation delivered by skilled healthcare professionals, but COVID-19 has put severe strain on healthcare systems. This study aims to explore whether digitally enabled, remotely supported rehabilitation for people with long COVID-19 can enable healthcare systems to provide high quality care to large numbers of patients within the available resources. Specific objectives are to (1) develop and refine a digital health intervention (DHI) that supports patient assessment, monitoring and remote rehabilitation; (2) develop implementation models that support sustainable deployment at scale; (3) evaluate the impact of the DHI on recovery trajectories and (4) identify and mitigate health inequalities due to the digital divide.
**Methods and analysis** Mixed-methods, theoretically informed, single-arm prospective study, combining methods drawn from engineering/computer science with those from biomedicine. There are four work packages (WP), one for each objective. WP1 focuses on identifying user requirements and iteratively developing the intervention to meet them; WP2 combines qualitative data from users with learning from implementation science and normalisation process theory, to promote adoption, scale-up, spread and sustainability of the intervention; WP3 uses quantitative demographic, clinical and resource use data collected by the DHI to determine illness trajectories and how these are affected by use of the DHI; while WP4 focuses on identifying and mitigating health inequalities and overarches the other three WPs.
**Ethics and dissemination** Ethical approval obtained from East Midlands – Derby Research Ethics Committee

### Strengths and limitations of this study

► Strengths include our interdisciplinary approach, which by combining methods common to engineering/computer science, for example, user-centred design and the human computer interaction lifecycle, with those from biomedicine, such as the Medical Research Council Complex Interventions Framework, promotes the likelihood of an intervention which is not only usable and acceptable, but also effective.

► Strong patient and public involvement input has helped define user requirements, including conceptualisation of long COVID-19 and focusing on function, fatigue and brain fog as well as ensuring the evaluation measures outcomes of interest to patients and clinicians.

► Intertwining service and research has enabled the rapid development and deployment of an evidence-based intervention, enabling treatment to reach patients rapidly during the pandemic.

► Our major limitation is the absence of a comparator group for the evaluation of effectiveness; our focus will be on profiling heterogeneity in recovery trajectories and conducting an exploratory mediation analysis to assess which intervention components and measures of app usage/engagement are associated with change in outcomes.

(reference 288199). Our dissemination strategy targets three audiences: (1) Policy makers, Health service managers and clinicians responsible for delivering long COVID-19 services; (2) patients and the public; (3) academics.

**Trial registration number** Research Registry number: researchregistry6173.

## BACKGROUND

Since the emergence of the new coronavirus, SARS-CoV-2 in Wuhan, China in December 2019[1] and the subsequent COVID-19 variants, it has become apparent that while many affected people make a full recovery, others are left with long-term disabling and distressing symptoms,[2] known as 'long COVID-19 syndrome'.[3 4] Prevalence estimates vary, but at least 50% of hospitalised[5 6] and 10% of non-hospitalised continue to have symptoms 12 weeks after the initial infection.[2 4] UK government figures suggest that by the end of July 2021, 5.9 million people in the UK had tested positive for COVID-19, of whom just under half a million (492 933) had been admitted to hospital (https://coronavirus.data.gov.uk/), suggesting that over 750 000 people in the UK may experience disabling symptoms persisting more than 12 weeks.

The pathophysiology of long COVID-19 has yet to be elucidated. It appears to be a multisystem disorder, characterised by variability in symptoms and a fluctuating trajectory,[4] which significantly impairs people's ability to work, look after their children, or engage with other activities. Despite variability in the nature and severity of reported symptoms, there are some core symptoms experienced by nearly all those with kong COVID-19: fatigue; cognitive impairment ('brain fog'); breathlessness; anxiety and depression.[2] These core symptoms are present in numerous other long-term conditions, and treated with well established, non-pharmacological interventions, including physiotherapy, nutritional advice, cognitive behavioural approaches, sleep hygiene and improving self-management skills. These interventions are most often delivered by Allied Health Professionals (AHP), such as physiotherapists, occupational therapists, clinical psychologists and dieticians, specialising in rehabilitation.

Prior to the COVID-19 pandemic, rehabilitation services were at capacity, and their ability to absorb over half a million additional patients requiring help has been further compromised by redeployment of many AHP to supporting acute services. Additionally, many services are system-focussed, so a patient with long COVID-19 who is breathless, fatigued and anxious could be directed to three different clinicians: a respiratory physiotherapist to help with breathing, a neurophysiotherapist or occupational therapist for fatigue, and a clinical psychologist for anxiety, making treatment extremely burdensome.[7]

One effect of the pandemic has been to rapidly accelerate the move toward the use of digital technologies to support healthcare.[8] Digitally supported rehabilitation, shown to be effective in other disease areas,[9–12] has the potential to address the challenge of providing timely healthcare to large numbers of individuals with a highly constrained workforce.[13] Digital health is not without its challenges: despite being shown to be effective in improving health outcomes, relatively few digital health interventions have made it into mainstream healthcare.[14 15] There are well recognised problems with implementation at scale; many DHI suffer from poor uptake or inadequate ongoing use,[16] reducing the likelihood of effectiveness. Furthermore, there are concerns around health inequalities and the digital divide, especially as COVID-19 disproportionately affects people from ethnic minority groups, socially deprived backgrounds and older people.[17] People from these backgrounds are also less likely to have access to digital technology,[18] so care delivered digitally risks exacerbating health inequalities. However, it has been shown that the digital divide can be overcome by careful design and implementation into routine healthcare.[19]

### Aims and objectives

The aim is to iteratively design, deploy and evaluate a digitally mediated, remote, supported rehabilitation programme for patients affected by COVID-19, which is used effectively, deployed at scale, and does not widen health inequalities.

Specific objectives are to:

1. Develop and refine a DHI that supports patient monitoring, remote rehabilitation and identifying patients needing further specialist investigation. To include:
   - Clinical pathways aligned with National Institute for Health and Care Excellence guidelines.[20]
   - A clinician-facing digital dashboard which displays data about individuals and selected cohorts.
   - A patient-facing mobile app to provide targeted, tailored rehabilitation according to patient symptoms. The app will collect patient-reported outcome measures (PROMs) and use intelligent algorithms and machine learning to promote engagement and tailor treatment advice under clinician guidance.
2. Determine implementation strategies to promote adoption, scale-up, spread and sustainability with a view to maximising population impact. To include assessing:
   - How best to integrate the DHI into clinical workflows, so it becomes fully normalised for healthcare professionals (HCPs), and patients experience seamless care.
   - The optimal amount of HCP input (time, skill set) required to support patients to engage effectively with the DHI, including those with low digital literacy, while managing large patient workloads safely.
   - Funding and commissioning models to promote long-term sustainability.
3. Assess the population impact of this model of care. To include:
   - Determining the reach (uptake and engagement) of the DHI, as a proportion of eligible patients and how this varies by demographic characteristics.
   - Exploring its impact on patient-reported outcomes and recovery trajectories.

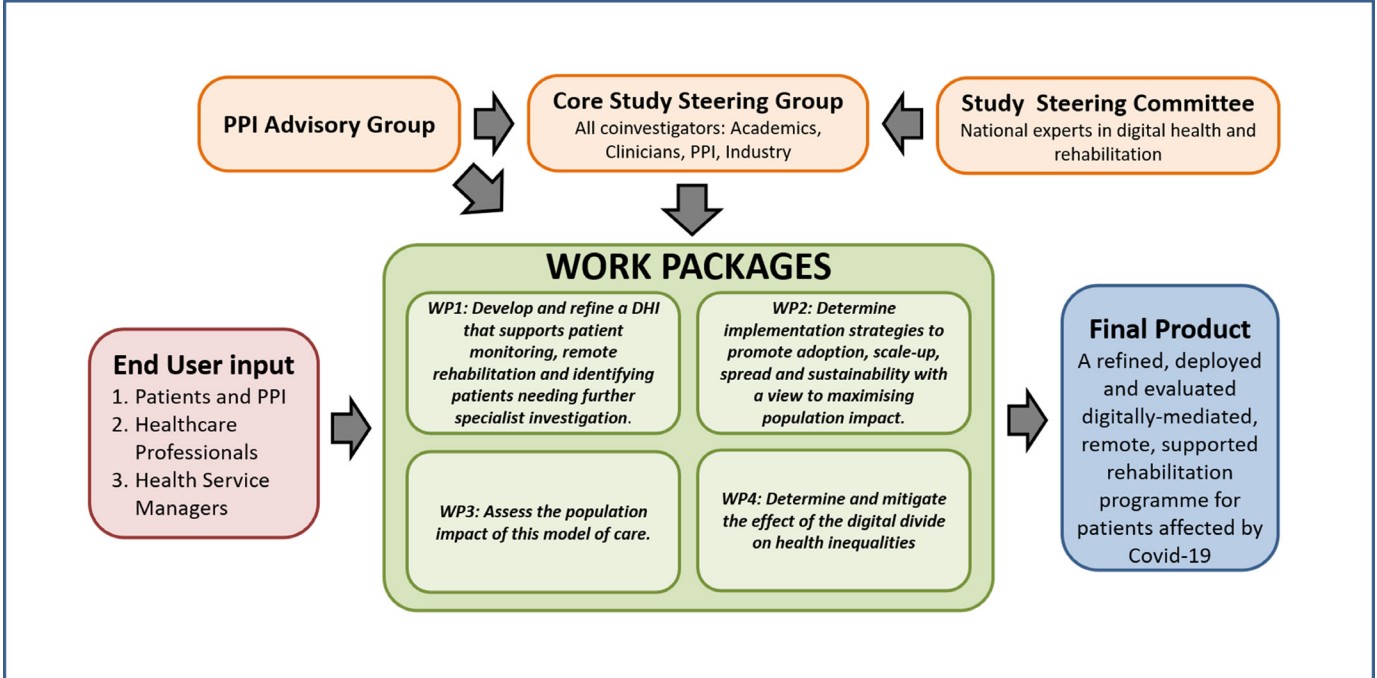

**Figure 1** Living With COVID-19 Recovery organisational structure. DHI, digital health intervention; PPI, patient and public involvement; WP, work packages.

– Assessing the cost implications to the National Health Service (NHS) of scaling-up and sustaining it.

4. Determine and mitigate the effect of the digital divide on health inequalities.
   – Identify patterns of differential uptake, use and apparent benefits of the DHI for people from ethnic minority or socially disadvantaged backgrounds.
   – Identify and test actions to mitigate any observed differential.

## METHODS
### Overall design
This project will combine research methods common to engineering and computer science (focused on developing a product that is safe, stable and meets user requirements) with those familiar to biomedical, behavioural and health service researchers (focused on effectiveness and population impact). Thus, it will follow the Medical Research Council (MRC) Framework for development and evaluation of complex interventions (phases 1, 2 and 4),[21] user-centred design (UCD) and the ISO 9241 Human-Computer Interaction (HCI) Lifecycle[22] for intervention developmentEvaluation will use mixed methods, combining qualitative and quantitative data. The work will be divided into four work-packages (WP) across 2 years, with each WP addressing an objective (figure 1).

### Key principles
All work undertaken will adhere to two key principles: (A) It addresses concurrently the aims of service and research, with the intention that the two should be mutually beneficial. The service priorities are to deliver a safe, effective and cost-effective service at speed in a resource-constrained environment; research requires us to generate new, generalisable knowledge. (B) Interdisciplinary approach with equal representation. The four communities represented in the team include: (1) patients, represented through patient and public involvement (PPI); (2) clinicians caring for patients with long COVID-19, including respiratory specialists, general practitioners and AHP; (3) industry, in the form of a small-medium enterprise specialising in digital health with substantial experience of deployment in the NHS and other healthcare systems; (4) academics. The academic disciplines represented in the team include computer science, HCI, digital health, behavioural science, social science and implementation research, statistics, health economics, health services research and clinical academics.

### Patient and public involvement
There is strong PPI representation throughout this study. The original application was developed with members of long COVID-19 social media groups, and the funding application was reviewed by PPI prior to submission. A PPI member (JB) is a co-investigator, named on the funding application, sits on the project steering committee and is a coauthor on this paper. Once funding was confirmed, we recruited additional PPI members resulting in two PPI on the steering committee, two PPI on each of the WP groups and a PPI advisory group. PPI members have contributed to: the overall design of the protocol; conceptualisation of long COVID-19, including prioritisation

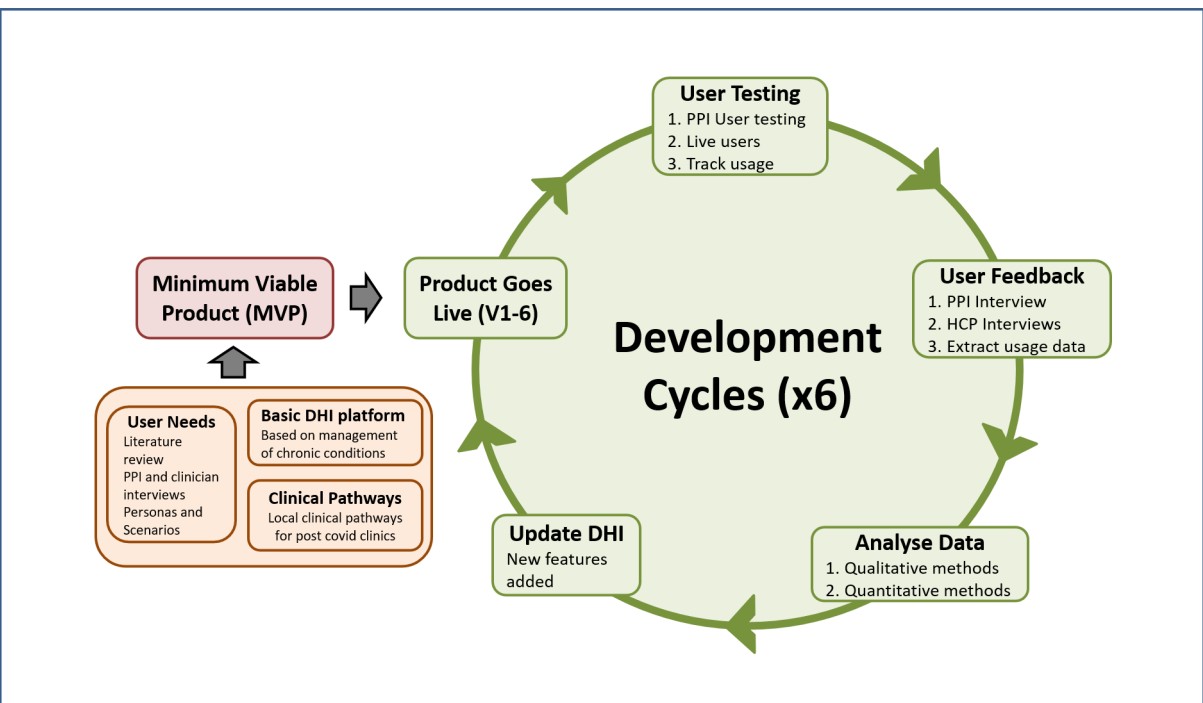

**Figure 2** Iterative cycles of development and refinement for Living With COVID-19 Recovery. DHI, digital health intervention; HCP, healthcare professional; PPI, patient and public involvement.

of the symptoms of fatigue and brain fog; writing and reviewing the content, navigation and functionality of the intervention; selecting outcome measures; data collection, particularly running focus groups with other PPI; data analysis, particularly around user requirements, and continue to contribute to the study conduct.

### Timeline

This is a 2-year project, that started in September 2020, with WP1 mainly focused on the first year of the project to design and optimise the product. WP3 will mainly be focused on the second year. WP2 and 4 will run throughout both years.

### WP1: development and refinement of the Living With COVID-19 Recovery programme

#### Design

Interdisciplinary team applying agile UCD and HCI methods to iteratively determine user requirements, to develop and refine the DHI by testing it against user requirements including optimising effectiveness in achieving desired behavioural changes (figure 2).

Inputs include: user requirements; emerging clinical evidence on Long COVID-19; evidence-based treatments; development of personas (an HCI technique involving developing rich, contextualised descriptions of a range of potential users[14]); qualitative user feedback and quantitative usage data. These data sources will be combined in an interative fashion, with six development cycles planned for the first year (figure 2), starting with an existing platform designed for use in other chronic health conditions such as cancer, rhumatoid arthritis and incontinence.

#### Participants

We have identified three main stakeholder groups for user requirements: patients with long COVID-19; HCPs caring for patients with long COVID-19, and health service managers responsible for long COVID-19 services. Initally, patient user requirements have been determined from PPI; recent ethics approval will allow data collection from patients using the Living With COVID-19 Recovery programme (LWCR) programme. HCPs and health service managers wil be identified initially through snowballing from the research team and subsequently from NHS teams using the service.

#### Data collection

We will use qualitative methods in real-time, including online meetings, interviews and focus groups with the three stakeholder groups, recorded with participant consent. Meetings will be driven by both research and service need to provide real-time insight into the needs of patients, health service managers and HCPs, as they adapt to the evolving pandemic, without imposing additional workload. Documents, including meeting agendas and minutes, and follow-up emails, will also be used as data sources.

Quantitative data: see WP3. For ethical reasons we are unable to contact or collect data directly from patients who have declined the LWCR service.

#### Data analysis

Recordings will be stored securely. Relevant sections will be transcribed by a general data protection regulation (GDPR) compliant transcribing company, anonymised

and checked for accuracy. Analysis will be done in two stages: first, a rapid, emergent analysis in parallel with data collection, focusing on actionable findings, fed immediately into the development process. Different data sources will be synthesised by the inter-disciplinary development team, through iterative discussions. Subsequent analysis will be in depth and inductive with transcripts coded by experienced qualitative researchers using NVivo, with additional deductive analysis where appropriate, driven by one of our theoretical frameworks or findings from the literature, critiqued in a multidisciplinary data clinic.

Quantitative data will be anonymised. Initial analyses will be descriptive, with subsequent, more complex analyses focusing on usage patterns with a view to maximising uptake and use.

### Outputs

Evidence-based DHI which is rapidly and fully integrated into routine care pathways of NHS long COVID-19 clinics nationally. The DHI will be routinely updated with new features for the App and Dashboard, such as questionnaires or content, every few months to increase usability, engagement and effectiveness based on participant feedback. Patients and clinicians will be made aware of changes through notifications on the DHI or emails and supported with a helpline.

### WP2: determine implementation strategies to promote adoption, scale-up, spread and sustainability with a view to maximising population impact

#### Design

Mixed-methods, using quantitative data to measure uptake and use at individual and clinic level, and qualitative data to explore and understand barriers and facilitators to implementation. Data collection, analysis and development of implementation strategies will be informed by normalisation process theory (NPT),[15 23] a sociological theory concerned with the work required to implement, embed and integrate (or 'normalise') new practices or technologies into routine healthcare.

#### Setting

The LWCR programme will be implemented in community long COVID-19 clinics nationally; these clinics will cover both urban and rural areas. The use of the LWCR programme is free for any NHS long COVID-19 clinic in England until September 2022. We conceptualise implementation as occuring at the macrolevel, mesolevel and microlevel. We define 'macro' as system level change, such as change across multiple local health economies at national or regional level, for example, NHS England (NHSE), Academic Health Science Networks (AHSNs)[24] or Integrated Care Systems (ICS).[25] 'Meso' refers to change across a single clinic or local health economy; and 'micro' refers to change at the level of the individual clinician or patient.

### Participants

Participants for this WP include anyone with responsibility for implementing change in the delivery of care for patients with long COVID-19 at any level. This includes staff working for NHSE or the Department of Health and Social Care (DHSC) with responsibility for policy and practice; commissioners of care; local health service leaders, such as leads or staff working for AHSNs or ICS, clinical and managerial leads working within or across health service Trusts (acute, primary or community care); clinicians responsible for specific clinics and/or delivering front line care to patients with long COVID-19, and patients with long COVID-19 or their carers. Participants also include professionals with particular relevant expertise needed to support the implementation of LWCR, such as understanding of information governance or procurement regulations in the NHS, and members of the research team with responsibility for setting up and delivering the service.

### Data collection

Data collection and analysis will be shared across WPs. Additional quantitative data collected for WP2 includes numbers of Trusts showing initial interest in deployment who decide against adoption, and demographic data on HCPs registered on the dashboard, including clinical specialty, level of experience, age, gender and ethnicity, collected through a proforma questionnaire completed by each clinic once only.

Additional qualitative data will include email trails and documents published by DHSC, NHSE, AHSNs, ICS or other health service bodies. Clinician feedback on the service will be collected as part of the cycles of iterative development and refinement. This service-related data may be augmented, where necessary, by specific interviews with selected individuals, for example, to obtain deeper insights into an issue raised in a meeting, or to test emerging interpretations of data. Where possible meetings will be recorded, with participant consent. Field notes and minutes will also form part of the dataset.

Data pertaining to individual uptake and use of the LWCR programme will include feedback from clinicians, particularly focusing on how they introduce the service to patients and steps taken to encourage uptake and use, particularly among patients who are less used to digital technology. Opt-in consent will be sought from patients and clinicians to record specific on-boarding consultations. Patients will be invited to participate in interviews with the research team to explore their experience of long COVID-19, the extent to which the service meets their perceived needs, and what could be done to improve the service for others.

### Data analysis

Analysis of quantitative data is described in WP3. Qualitative data will be analysed in two stages, as described for WP1, with the first stage focusing on actionable findings to promote deployment and the second focusing

on transferable learning about implementation of DHI. In this second stage, data will be analysed inductively, with data-driven codes subsequently mapped onto NPT concepts, looking specifically for disconfirming data or data that does not map onto NPT.

### Outputs

Outputs from WP2 will include:
► A fully developed implementation package, detailing the process of successful implementation of a DHI into routine healthcare in a range of healthcare setting (primary, secondary and community care).
► A sustainable, scalable business model, allowing for ongoing maintenance and development of the product while it is used at scale.
► Transferable learning about optimising the integration of digital health into routine care, ensuring optimal use by HCPs and patients, including an understanding of how implementation models can mitigate the digital divide.

### WP3: assess the population impact of this model of care
### Design

Quantitative workpackage. Population impact is defined as reach multiplied by effect.

### Participants

The target for year 1 will be to recruit 1000 patients registering with the DHI. This will enable us to estimate the overall proportion of patients engaging with the app with a high-level of precision (maximum width of 95% CI is ±3.2%). Similar or increased levels of precision will be achieved when estimating the proportion of those referred who register with the intervention. The sample size in year 2 has not been prespecified and will depend on practical constraints imposed by the scale of the pandemic as the digital intervention is rolled out nationally. Power calculations will be carried out for the year 2 analyses based on the 'pilot' data from year 1. This will include power for the proposed quasi-experimental analysis of the effect of the DHI on outcomes at 3-month and 6-month follow-up.

### Data collection

Quantitative data on numbers of patients, HCPs and long COVID-19 clinics registered to use LWCR. Additionally, patient demographic data, clinical and service data and app usage patterns, as recorded through the LWCR platform will be provided.

Demographic data will be obtained on patients' age, gender, ethnicity, highest level of educational attainment and socioeconomic status (using Index of Multiple Deprivation, calculated from the postcode) when patients first register on the app. This data will be entered by HCPs or the patients themselves.

Clinical data will be obtained from patient-completed PROMs on the app. Patients will be asked to complete all the PROMs at baseline to help with clinical assessment, and thereafter, only those PROMs which the clinician and

patient deem to be clinically relevant and helpful for clinical management.

Validated PROMs include: the Work and Social Adjustment Scale which measures functional impairment and is our primary outcome measure[26]; EuroQol 5 dimensions, 5 response level instrument (EQ-5D-5L) for health related quality of life[27]; Functional Assessment of Chronic Illness Therapy - Fatigue (FACIT-F) for fatigue[28]; Generalised Anxiety Disorder Assessment (GAD-7) for anxiety[29]; Personal Health Questionnaire Depression Scale (PHQ-8) for depression[30]; Medical Research Council (MRC) dyspnoea scale[22] and Dyspnoea-12[31] for breathlessness, and Perceived Deficits Questionnaire-5 item (PDQ-5) for cognitive impairment.[32] Additional questions explore levels of physical activity and overall recovery from COVID-19.

Service use data will be obtained when patients complete a service use questionnaire (every 4 weeks), detailing the number of healthcare appointments, days off work and days as a hospital inpatient they have experienced.

Usage data including the date and time of each page view by each patient are recorded automatically by the programme.

### Data analysis

Each patient will be assigned a unique participant identification number. The linking key will be held by Living With, as data processors for the Clinical Trusts. Pseudonymised data, where date of birth is replaced by age and postcode replaced with IMD, will be shared with the research team. Demographic characteristics and baseline symptoms of the registered 'treatment seeking' population of long COVID-19 sufferers will be profiled using descriptive statistics.

### *Usage and engagement*

Patient engagement with the app will be reported using a stratification of patients into low, medium and high categories based on number of logins over the first 12 weeks. Demographic and baseline clinical factors influencing uptake and use will be identified through logistic regression, with multivariate methods used to model patterns of app engagement over time.

### *Recovery trajectories*

Generalised linear mixed models (GLMMs) and latent class trajectory modelling will be used to examine stability and change in recovery trajectories for the longitudinal PROMs data, and to identify homogeneous clusters of patients with contrasting clinical phenotypes. Time since registration with the programme will initially be included as a linear covariate, but we will explore the functional form of the effect of time to allow for phenomena such as a plateau in symptom recovery or a period of sudden rapid improvement or relapse. Models will include random slopes and intercepts, allowing for individualised differences in severity of baseline symptoms and subsequent trajectories. Fixed effects will

include age, gender, ethnicity, IMD and educational attainment.

### Mediation analysis of intervention components

A mediation analysis will be undertaken in which intervention components and measures of patient app usage and engagement (number of logins and assessments completed) will be added to longitudinal models as explanatory factors. We will explore whether increased engagement by clinicians, such as messages sent and reviews of patient assessments on the dashboard, is linked with improved patient outcomes. We will assess whether there is a dose–response relationship between usage of the app/introduction of new content and change in outcomes.

### Effectiveness of the LWCR programme

We will investigate the feasibility of creating a synthetic comparator arm from patients with long COVID-19 enrolled in suitable comparator cohorts, hosted in BREATHE-SAIL.[33] Direct comparisons will be made between patients receiving the evidence-based interventions and the comparator group(s) in multivariable GLMMs with propensity score methods used to provide control for measured confounders.

### Equality

The acceptability of the intervention for people from disadvantaged backgrounds will be explored through logistic regression analysis of these factors as predictors of recruitment, usage and retention. Differential effects of the DHI on outcomes for disadvantaged groups will be investigated through inclusion of interactions between intervention uptake and ethnicity, age, educational status and IMD in the GLMMs.

### Missing data

Sensitivity analyses will be conducted to examine the influence of missing data on the key study findings. This will include the use of multiple imputation methods, where the assumption that the data are missing at random is considered appropriate, and tipping point analysis[34] to assess sensitivity of the results to data that are missing not at random.

### Health economic evaluation

The health economic evaluation will estimate the cost of delivering the DHI to the target population, the cost consequences of the DHI for the NHS services compared with standard care, and a budget impact analysis over the next 3–5 years[35] that assesses the costs and potential savings resulting from commissioning the proposed DHI. To generate costs from the resource use data, appropriate unit costs,for example, price lists from the app developer, to cost items related to app development, and Unit Costs for Health and Social Care[36] to cost staff time, will be applied.

Relevant costs will be identified by examining how the DHI impacts the patient, hospital and the healthcare system. These will include:

► Resources associated with the development and hosting of the patient-facing app.
► Resources incurred with the scaling up of the DHI, such as maintenance, updating, data storage, implementation and training.
► Resource use associated with HCP support of patients using the service.
► Impact on health service use.

### Outputs

Outputs from WP3 will include detailed understanding of patient symptoms and illness trajectories over time, estimates of the impact of the digital health programme on these trajectories, and an understanding of the relative cost-effectiveness of the LWCR programme compared with standard care, that is, the care provided to those patients not supported by the LWCR programme.

## WP4: determine and mitigate the effect of the digital divide on health inequalities

### Design

This cross-cutting WP will ensure that the work undertaken in the other three WPs is fully sensitive to the needs of those who might be at risk of exclusion, such as minority ethnic groups, socially disadvantaged or older people. It will seek to identify evidence of the digital divide and health inequalities, and work to mitigate these. We will undertake a systematic scoping review to identify features of DHI design and deployment that enable engagement by people with low digital/health literacy, for example, key content is presented using graphics, video, and audio to enhance accessibility.

In relation to WP1, the focus will be on ensuring that the personas used in developmental design include minority ethnic, socially disadvantaged and older people, to improve usability and engagement in these groups. We will ensure the content is written for a reading age of entry level 3 (attained by over 80% of the UK population)[37] and therefore, understandable by most patients. Design features will allow family or carers to log in and support patients. The systematic scoping review will inform intervention design and deployment to maximise engagement by people with low digital/health literacy.

In relation to WP2, the focus will be on identifying and exploring factors in the implementation process that could hinder uptake and increase health inequalities, or conversely, help to mitigate inequalities. For example, translation of key content into different languages, the use of digital champions to help support patients, referral to charities that supply smart phones and advise how to access low-cost data or free Wi-Fi, to help facilitate increased access to all groups.

In relation to WP3, the focus will be on ensuring the analysis of the outcome measures can be compared across different demographic groups so that disparities can be identified, and exploring the factors that can influence differential uptake, use, and impact across the different clinics and demographic groups.

## Outputs

An intervention which is accessible to people from a range of demographic backgrounds and that collects/analyses/acts on data suggesting differential uptake or impact of the DHI by people from these demographic groups; and an understanding of the features of the implementation model which promote participation by people from ethnic minority, disadvantaged or older groups.

## Ethics and dissemination

This protocol was approved by the East Midlands – Derby Research Ethics Committee (reference 288199). Ethical consideration was given to obtaining patient consent and maintaining patient anonimity. First consent to contact patients by email will be obtained using a tick box on the App. Only those patients that opt in will be contacted and asked if they wish to participate in interviews. All patient data collected from the LWCR product will be psudeoanonymised by Living With. Patient email addresses from those who agree to be contacted will be stored separately in the university data safe haven in accordance with the GDPR and data governance protocols.

Our dissemination strategy targets three audiences: (1) policy-makers, health service managers and clinicians responsible for delivering long COVID-19 services; (2) patients and the public and (3) academics. Findings will be disseminated at conferences and papers to target academics, through nationally advertised workshops for clinicians and administrators working in or running long COVID-19 clinics, and through news and social media for patients.

## DISCUSSION

This protocol describes the processes for developing, deploying and evaluating digitally enabled remote, supported rehabilitation for people with long COVID-19 syndrome. Strengths include the inter-disciplinary approach, with a partnership between patients, clinicians, industry and academics, plus combining approaches used in engineering/computer science, for example, UCD and the HCI lifecycle, with those more familiar to biomedical researchers, for example, the MRC Framework for complex interventions. The strong use of theory to underpin the research should enhance transferability of findings. Putting equal priority on service needs and research is innovative, but brings challenges. These include relying on diversity of service delivery for obtaining an appropriate comparator and the reliance on data obtained for clinical purposes for the impact evaluation. We expect there to be substantial amounts of missing data, particularly in the PROMs, as clinicians and patients are likely to focus on recording symptoms of most interest to them or suffer from questionnaire fatigue. However, no one study can answer all relevant research questions about an innovation, and we agree with the argument that a DHI should achieve a reasonable level of stability before being evaluated in a randomised controlled trial.[38] This initial study will provide considerable insight into patient, clinician and health service requirements for digitally supported rehabilitation for long COVID-19,

together with very substantial, real world data on acceptability, uptake and use, going far beyond the usual pilot studies so often used as evidence of acceptability and feasibility, with data on scalable, and hopefully, sustainable, deployment. It will also generate data on symptoms and illness trajectories in a treatment-seeking population which includes non-hospitalised patients, making it unique among current cohort studies. Long-term investment will require causal evidence of effectiveness, and NIHR has funded the STIMULATE-ICP project, which includes a cluster randomised controlled trial comparing LWCR to standard care,[39] due to report in 2023. Other future work will consider whether a digital tool could help stratify long COVID-19 patients by their clinical severity[40] or combine with other digital help products such as telemedicine to achieve better outcomes.[41]

We believe that our approach could help alleviate the well known delay in translating research findings into practice[42] while simultaneously promoting the sustainable, scalable adoption of evidence-based interventions and adding to the research base in digital health.

**Author affiliations**
[1]Research Department of Primary Care and Population Health, University College London, London, UK
[2]Department of Primary Care and Population Health, University College London, London, UK
[3]UCLIC, University College London, London, UK
[4]Psychology, University of Southampton, Southampton, UK
[5]Computer Science, University College London, London, UK
[6]Department for Applied Health Research, University College London, London, UK
[7]Respiratory Medicine, University College London Hospitals NHS Foundation Trust, London, UK
[8]Institute of Health Research, University of Exeter Medical School, Exeter, UK
[9]Academic Unit of Respiratory Medicine, University College London, London, UK
[10]Department of Respiratory Medicine, Barts Health NHS Trust, London, UK
[11]Camden and Islington NHS Foundation Trust, London, UK
[12]Living With, London, UK

**Contributors** All authors (EM, HG, JB, AB, KB, DF-R, MG, FLH, MH, WH, JRH, HH, SL, PP, WR, CR, RS, FAS and JW), with the exception of TC and SW, contributed to the development of the successful funding application on which this protocol is based. TC further developed and operationalised the protocol as part of the ethics submission and SW elaborated the statistical analyses. EM and HG are cochief investigators and JB, AB, KB, DF-R, MG, FLH, MH, WH, JRH, HH, SL, PP, WR, CR, RS, FAS and JW are coinvestigators on the project. EM wrote the first draft of the paper and incorporated comments on the draft manuscript from all authors. HG wrote the response to the reviewer comments after discussion and comments from all authors. All authors have read and approved the final version.

**Funding** This study is funded by the National Institute for Health Research (NIHR) Crossprogramme (HS&DR) COVID-19 (project reference NIHR132243 - Supported remote rehabilitation post COVID-19: development, deployment and evaluation of a digitally enabled rehabilitation programme).

**Disclaimer** The views expressed are those of the author(s) and not necessarily those of the NIHR or the Department of Health and Social Care.This report is independent research supported by the National Institute for Health Research ARC North Thames.

**Competing interests** CR and JW are directors of the provider company, Living With, which has a suite of digital health interventions used in healthcare systems including the NHS.

**Patient and public involvement** Patients and/or the public were involved in the design, or conduct, or reporting, or dissemination plans of this research. Refer to the Methods section for further details.

**Patient consent for publication** Not applicable.

**Provenance and peer review** Not commissioned; externally peer reviewed.

**ORCID iDs**
Elizabeth Murray http://orcid.org/0000-0002-8932-3695
Henry Goodfellow http://orcid.org/0000-0002-0979-2839
Ann Blandford http://orcid.org/0000-0002-3198-7122
Delmiro Fernandez-Reyes http://orcid.org/0000-0001-5070-9198
John R Hurst http://orcid.org/0000-0002-7246-6040
William Ricketts http://orcid.org/0000-0002-0475-0744

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
