## [Reviewer comments · BMJ Open]

ARTICLE DETAILS

TITLE (PROVISIONAL)	Development, deployment and evaluation of digitally-enabled, remote, supported rehabilitation for people with Long Covid (Living With Covid Recovery). Protocol for a mixed methods study.
AUTHORS	Murray, Elizabeth; Goodfellow, Henry; Bindman, Julia; Blandford, Ann; Bradbury, Katherine; Chaudhry, Tahreem; Fernandez-Reyes, Delmiro; Gomes, Manuel; Hamilton, Fiona; Heightman, Melissa; Henley, W; Hurst, John; Hylton, Hannah; Linke, Stuart; Pfeffer, Paul; Ricketts, William; Robson, Chris; Singh, Richa; Stevenson, Fiona; Walker, Sarah; Waywell, Jonathan

VERSION 1 – REVIEW

REVIEWER	Yan, Zhipeng The University of Hong Kong
REVIEW RETURNED	30-Sep-2021

GENERAL COMMENTS	This will be an impactful study for monitoring and treatment of Long COVID-19 patients, despite it is a single-arm prospective study. It uses digital interventions to monitor the progress of the patient. Undoubtedly, digital intervention is cost-effective and may be the future trend of rehabilitation & other chronic diseases due to the huge patient size. Thus, the study is a good idea to fill in the research gap on how effective digital interventions will be in this regard. Meanwhile, the key to its success will be resources reallocations to prioritize the needs of patients, validated reporting systems/scales for the disease, and subsequent rehabilitation resources available. Meanwhile, there are few points I would like the authors to clarify: 1. Recommendation on stratifications of Long COVID-19 patients by their clinical severity. The previous study has shown that despite having Long COVID-19 Syndrome, they are with different severity; thus, different rehabilitation needs in terms of frequency of clinical visits, schedule of rehabilitations etc (Yan, 2021)... Therefore, the study can consider stratifying users into different groups based on their baseline clinical severity, according to clinical symptoms reporting, laboratory data/imaging data (if any). The digital interventions will be useful for monitoring the clinical severity of all patients, in particular those of mild to moderate severity. It can also document the treatment trajectories/disease progression (if any). References: Yan, Z.; Yang, M.; Lai, C.-L., Long COVID-19 Syndrome: A Comprehensive Review of Its Effect on Various Organ Systems and Recommendation on Rehabilitation Plans. Biomedicines 2021, 9 (8), 966. 2. Digital interventions should serve a dual purpose, both monitoring and therapeutic effects. Considerations include online consultations, rehabilitation exercises videos, tracking of rehabilitations follow-up progress and bookings, and common Q&A / forum sessions for patient support groups. In view of the emergence of telemedicine
--

(Kichloo, 2020), the use of digital interventions for both monitoring and long-term management purpose may be a future research direction. i.e. effectiveness of digital interventions in improving their clinical severity score, in particular, Long COVID-19 syndrome?

References:

Kichloo A, Albosta M, Dettloff K, et al. Telemedicine, the current COVID-19 pandemic and the future: a narrative review and perspectives moving forward in the USA. *Fam Med Community Health*. 2020;8(3):e000530. doi:10.1136/fmch-2020-000530

3. Digital Intervention should be incorporated into a community-based approach, with community rehabilitation staff being in charge of few cases for face-to-face interventions (if necessary). Patients with mild to moderate severity with suspicious features should be followed up by community rehabilitation staff. This is more cost-effective and can be considered in the future. This can also address your WP4 problem for those missed up population without access to digital technology: eg ethnic minority groups, socially deprived backgrounds and older people. Under your research plan, will there be outreaches?

4. in line 52 of p5 on the “Methods – key principles” part, this digital intervention is seen to be cost-effective; while in line 28 of p.10 on “WP3- Health economic evaluation”, the cost will be compared to standard care. Can you clarify what is standard care you are referring to? Patients may be of different severity and how can they be matched to each other? What scales are you using to match up comparable populations for cost comparison?

5. On P.9 line 19-24 talking about “PROMS” measurement, you mentioned the use of Work and Social Adjustment Scale (WSAS), EQ-5D-5L, FACIT-F, GAD-7, PHQ-8, MRC Dyspnea scale, Dyspnea-12, and PDQ-5 for a comprehensive assessment of Long COVID-19 patients. Due to the numerous scale measurements, you are going to use, one of the limitations will be substantial amounts of missing data particularly in the PROMS, as mentioned on p.11 line 34 “Discussion part”. Use of lots of parameters for measurement despite gives more accurate evaluations, but it’s more tedious to data-collectors and running a higher risk of missing data due to refusal or cooperation by users. There are 3 questions:

1) have you considered simplifying the scale of measurements since it involves many parameters for measurement? Is there a simpler screening tool for measurement?
2). Any evidence to support why you specifically chose these PROMS parameters? Recommend quoting the common clinical presentations of Long COVID-19 syndrome to justify your choice, and with reference to compare why not using currently existing tools?

3) any other tools that you can consider using e.g Yorkshire rehabilitation screening tool

(<https://ipswichandeastsuffolkccg.nhs.uk/LinkClick.aspx?fileticket=B21BKxLNIpo%3D&portalid=1>) or Newcastle Post-COVID syndrome Follow up Screening Questionnaire?

If your team has decided not to use currently existing screening tools, can you explain the reason? It is absolutely fine to use other validated tools / develop a new tool for assessment, but this may be a question of interest to readers to justify your choice of assessment tools.

6. Minor typos e.g. abstract introduction part “to rehabilitation delivered by skilled healthcare professional), but Covid-19...”-> the bracket after "professional" and pleural form of professional, article summary part 2nd point “evaluation measures outcomes of interest to patients ...” -> pleural form of interest = interests etc.

Overall, this is a good research protocol that will fill in our research gap on the effectiveness of telemedicine / digital interventions for monitoring & managing chronic diseases (Long COVID-19 Syndrome).

REVIEWER	Churchill, Katie Alberta Health Services, Allied Health Professional Practice & Education
REVIEW RETURNED	11-Oct-2021

GENERAL COMMENTS	Thank you for the opportunity to review the protocol for the study titled "Development, deployment and evaluation of digitally-enabled, remote, supported rehabilitation for people with Long COVID (Living with COVID Recovery). Protocol for a mixed methods study". The proposed study will make a significant contribution to the field of rehabilitation for Long Covid and the publication of the protocol will facilitate future work for other researchers, decision makers and health care providers. Please see attached letter for feedback and minor suggested revisions.
---

VERSION 1 – AUTHOR RESPONSE

Reviewer 1:

1. Recommendation on stratifications of Long COVID-19 patients by their clinical severity. The previous study has shown that despite having Long COVID-19 Syndrome, they are with different severity; thus, different rehabilitation needs in terms of frequency of clinical visits, schedule of rehabilitations etc (Yan, 2021)... Therefore, the study can consider stratifying users into different groups based on their baseline clinical severity, according to clinical symptoms reporting, laboratory data/imaging data (if any). The digital interventions will be useful for monitoring the clinical severity of all patients, in particular those of mild to moderate severity. It can also document the treatment trajectories/disease progression (if any).

Yan, Z.; Yang, M.; Lai, C.-L., Long COVID-19 Syndrome: A Comprehensive Review of Its Effect on Various Organ Systems and Recommendation on Rehabilitation Plans. *Biomedicines* 2021, 9 (8), 966.

Unfortunately, the digital product doesn't categorise long covid patients into different levels of severity. We decided early on that we didn't want the product to be able to stratify patients as we felt it was more appropriate for the clinicians to do this. This was especially the case in Long Covid where the knowledge base was weak and constantly changing. Our model of care is that the product function is to provide information only to the clinician who can then stratify the patients accordingly based on data from the platform and other sources such as bloods. The digital product itself doesn't have access to recent investigations or blood test which were used in the paper referenced above to accurately stratify patients. Also, for the digital product to stratify we would need to validate each category of Long covid for the questionnaires we used, and this is outside the scope of this project and no known categories of long covid were known when we started this project.

2. Digital interventions should serve a dual purpose, both monitoring and therapeutic effects. Considerations include online consultations, rehabilitation exercises videos, tracking of rehabilitations follow-up progress and bookings, and common Q&A / forum sessions for patient support groups. In view of the emergence of telemedicine (Kichloo, 2020), the use of digital interventions for both monitoring and long-term management purpose may be a future research direction. i.e. effectiveness of digital interventions in improving their clinical severity score, in particular, Long COVID-19 syndrome?

Kichloo A, Albosta M, Dettloff K, et al. Telemedicine, the current COVID-19 pandemic and the future: a narrative review and perspectives moving forward in the USA. *Fam Med Community Health*. 2020;8(3):e000530. doi:10.1136/fmch-2020-000530

Yes, the aim of our product is to support both monitoring and long term management. You are correct that the long-term goal is to determine clinical effectiveness. This will be done in a recently funded clustered RCT called Stimulate ICP run by another research team. See link for more information (<https://www.arc-nt.nihr.ac.uk/research/projects/stimulate-icp-improving-diagnosis-treatment-and-care-of-long-covid/>). This project is mentioned in the discussion section.

3. Digital Intervention should be incorporated into a community-based approach, with community rehabilitation staff being in charge of few cases for face-to-face interventions (if necessary). Patients with mild to moderate severity with suspicious features should be followed up by community rehabilitation staff. This is more cost-effective and can be considered in the future. This can also address your WP4 problem for those missed up population without access to digital technology: eg ethnic minority groups, socially deprived backgrounds and older people.

Under your research plan, will there be outreaches?

We agree with your observation and NHS long covid clinics are in the community. Our digital product has been designed to be used in a community-based or outreach clinic as you describe.

We have edited WP2 methods section

“The LWCR programme will be implemented in community Long Covid clinics nationally, these clinics will cover both urban and rural areas.”

4. in line 52 of p5 on the “Methods – key principles” part, this digital intervention is seen to be cost-effective; while in line 28 of p.10 on “WP3- Health economic evaluation”, the cost will be compared to standard care. Can you clarify what is standard care you are referring to? Patients may be of different severity and how can they be matched to each other? What scales are you using to match up comparable populations for cost comparison?

The methods used in the cost comparison, including the definition of ‘comparator group’, will be similar to those used for the effectiveness analysis. Standard care will be defined as the care provided to the patients not supported by the App within each Long Covid clinic. We will examine the cost trajectories for patients exposed to the DHI compared to that of patients not exposed to the DHI (comparator group). The methods used to match up the comparison groups, i.e. propensity score matching, will be the same as described in the data analysis section.

Edited manuscript to explain comparator group more clearly.

“i.e the care provided to those patients not supported by the LWCR programme”

5. On P.9 line 19-24 talking about “PROMS” measurement, you mentioned the use of Work and Social Adjustment Scale (WSAS), EQ-5D-5L, FACIT-F, GAD-7, PHQ-8. MRC Dyspnea scale, Dyspnea-12, and PDQ-5 for a comprehensive assessment of Long COVID-19 patients. Due to the numerous scale measurements, you are going to use, one of the limitations will be substantial amounts of missing data particularly in the PROMS, as mentioned on p.11 line 34 “Discussion part”. Use of lots of parameters for measurement despite gives more accurate evaluations, but it’s more tedious to data-collectors and running a higher risk of missing data due to refusal or cooperation by users. There are 3 questions:

1) have you considered simplifying the scale of measurements since it involves many parameters for measurement? Is there a simpler screening tool for measurement?

2). Any evidence to support why you specifically chose these PROMS parameters? Recommend quoting the common clinical presentations of Long COVID-19 syndrome to justify your choice, and with reference to compare why not using currently existing tools?

3) any other tools that you can consider using e.g Yorkshire rehabilitation screening tool (<https://ipswichandeastsuffolkccg.nhs.uk/LinkClick.aspx?fileticket=B2IBKxLNlpo%3D&portalid=1>) or Newcastle Post-COVID syndrome Follow up Screening Questionnaire?

If your team has decided not to use currently existing screening tools, can you explain the reason? It is absolutely fine to use other validated tools / develop a new tool for assessment, but this may be a question of interest to readers to justify your choice of assessment tools.

We very carefully considered the measurements (questionnaires) we used and were directed by patient and clinician need. As the digital App is being used as part of the NHS service we required already validated questionnaires for safety reasons that would allow clinicians to be able to easily interpret and compare scores from other diseases. Each questionnaire used needed to have a research paper validating its use and they are referenced in the paper. At the time of this digital product going live (August 2020) the Yorkshire and Newcastle rehabilitation screen tools were not considered to be validated for screening and monitoring of the core symptoms of Long Covid (fatigue, anxiety, brain fog etc) and therefore were not used. If this changes or the tool is mandated by the NHS then we will consider incorporating it. It would, however, be very difficult to change questionnaires once they have started being used as it would be very confusing for clinicians and patients and challenging to analyse so this would require a strong case for change.

The number of questionnaires used was due to the wide range of symptoms that Long Covid causes. We are aware of the risk of questionnaire fatigue and as such the App encourages you to only complete questionnaires once for all the symptoms and then only for those symptoms that are most important to the patient.

6. Minor typos e.g. abstract introduction part “to rehabilitation delivered by skilled healthcare professional), but Covid-19...”-> the bracket after "professional" and plural form of professional, article summary part 2nd point “evaluation measures outcomes of interest to patients ...” -> plural form of interest = interests etc.

Thank for your feedback. We can confirm the manuscript has now been carefully proofread.

Reviewer 2

General Feedback

The protocol discusses the use of Implementation Science and Normalisation Process Theory. The protocol would be strengthened if the authors provide detail on which Implementation Science theory or framework will be used to support the adoption, scale up and spread of the intervention.

Thank you for this comment. We have discussed it as a team and the implementation theory we will use is Normalisation Process theory and we have edited the text to reflect this more clearly.

“Data collection, analysis and development of implementation strategies will be informed by Normalisation Process Theory (NPT) (15) (22), a sociological theory, about the work required to implement, embed and integrate (or “normalise”) new practices or technologies into routine healthcare.”

As well, the journal indicates that timelines should be outlined in the protocol but timelines are currently not present.

We have added timelines section in the methods

“Timeline. This is a two year project with WP1 mainly focused on the first year of the project to design and optimise the product. WP3 will mainly be focused on the second year. WP2 & 4 will run throughout both years.”

There are a lot of acronyms in this protocol, please consider using full terms if the word count permits to increase flow/readability

We agree the use of full text terms aids readability and will leave their inclusion to the discretion of the BMJ open editor due to the impact on the wordcount.

As well, it would be helpful to include more information about the context of where this will be implemented. It is urban or rural? What type of funding model supports access to the Living With Long COVID Recovery programme? More specific contextual information will help with readability and generalizability for other health systems to learn from this work.

This has been added to WP2

“The LWCR programme will be implemented in community Long Covid clinics nationally, these clinics will cover both urban and rural areas. The use of the LWCR programme is free for any NHS long covid clinic whilst the research project is ongoing. After the project has finished trusts will need to pay to continue to use the product.”

Page 2, Line 13: remove extra parenthesis

This has been edited

Page 2, Line 28: missing Science after engineering/computer. Remove extra parenthesis after biomedicine. Consider capitalizing Work Packages if WP acronym going to be used in abstract.

This has been edited

Page 2, Line 35: if space permits, consider adding more details about WP4 as the focus on determining and mitigating the impact of digital divide on health inequities is a strength of this protocol and should be highlight upfront in the abstract

Added to the abstract “while WP4 focuses on identifying and mitigating health inequalities and overarches the other three WPs.”

Please see below for further updates to the WP4 section.

Page 3, Line 15: consider using full term for Patient and Public Involvement (PPI) as that acronym is not previously used in this section.

This has been edited

Page 6, Line 31: the design of work package 1 is very technical and uses two acronyms not previously identified in the paper (UCD, HCI)

The terms have been written in full to aid with the understanding.

Added “User Centred Design (UCD) and Human Computer Interaction (HCI)”

Page 6, Line 40: can you provide more information about the existing platform that was designed for other populations. What features make it a good fit for Long COVID?

Added “chronic health conditions such as cancer, rheumatoid arthritis and incontinence”

Page 6, Line 44: acronym HCP not used prior, use full term health care provider if word count permits to cut down on acronyms.

The term have been written in full to aid with the understanding

“Health Care Provider (HCP)”

Page 7, line 21 - it states the DHI will be routinely updated to increase usability, engagement and effectiveness. What kind of updates? Will these updates only impact users or will clinical practice changes need to be implemented to support these changes? How would this be supported?

I have added new information to the text to answer these questions. The changes will impact on both patients and clinicians but any changes to the dashboard that clinicians use will be guided by their feedback and designed to minimise any disruption. Living With have a support number that patients and clinicians can call if they have any issues using the product.

Added “The DHI will be routinely updated with new features for the App and Dashboard such as questionnaires or content every few months to increase usability, engagement and effectiveness based on participant feedback. Patients and Clinicians will be made aware of changes through notifications on the DHI or emails and supported with a helpline.”

Page 7, Line 35 – it is strongly recommended that the study team pick an implementation science model or theory and outline how it will be used in this protocol.

We have made it clearer in the text that the implementation theory is Normalisation Process theory.

“Data collection, analysis and development of implementation strategies will be informed by Normalisation Process Theory (NPT) (15) (22), a sociological theory, about the work required to implement, embed and integrate (or “normalise”) new practices or technologies into routine healthcare.”

Page 10, Line 25 - Health Economic Evaluation: The study team will be using the EQ-5D-5L, was there consideration on calculating Quality Adjusted Life Years (QALYS) to support the economic evaluation?

Yes, QALYs will be calculated. The primary analysis will take the form of cost-consequences analysis, in which differences in costs and benefits between the DHI and the comparator group will be assessed, but not necessarily combined into a single metric, such as cost per QALY assessments. Page 10, Work Package 4: Determine and Mitigate effects on health inequities. This is a very important part of the protocol yet there is limited detail provided about this work package. If word count permits, consider elaborating on this. Will a particular theory guide this work? How will language barriers be addressed? How will people without access to the digital health intervention get access to this 3 technology? Perhaps these are bigger issues than what can be addressed in this study but it's important that they are acknowledged and identified for future research and learning across organizations.

We did not a priori use a specific conceptual framework for WP4; we wanted to wait to see if a relevant conceptual framework or theory is identified from the systematic scoping review (SSR), detailed below and currently underway. If so, we will determine how well themes from interviews with patients map to this theory, but otherwise we will use the results of both SSR and interviews to inform development of a conceptual framework.

Regarding the reviewer's points about language barriers and improving access to the DHI for people affected by the digital divide, we have added more detail about these aspects to the section on WP4, below, in relation to WP2:

Edited section on

Work Package 4: Determine and mitigate the effect of the digital divide on health inequalities.

Design. This cross-cutting WP will ensure that the work undertaken in the other three WPs is fully sensitive to the needs of those who might be at risk of exclusion, such as minority ethnic groups, socially disadvantaged or older people. It will seek to identify evidence of the digital divide and health inequalities, and work to mitigate these. We will undertake a systematic scoping review to identify features of DHI design and deployment that enable engagement by people with low digital/health literacy, e.g. key content is presented using graphics, video, and audio to enhance accessibility.

In relation to WP1 the focus will be on ensuring that the personas used in developmental design include minority ethnic, socially disadvantaged, and older people, to improve usability and engagement in these groups. We will ensure the content is written for a reading age of entry level 3 (attained by over 80% of the UK population) (34) and therefore understandable by most patients. Design features will allow family or carers to log in and support patients. The systematic scoping review will inform intervention design and deployment to maximise engagement by people with low digital/health literacy.

In relation to WP2 the focus will be on identifying and exploring factors in the implementation process that could hinder uptake and increase health inequalities, or conversely, help to mitigate inequalities. For example, translation of key content into different languages, the use of digital champions to help support patients, referral to charities that supply smart phones and advise how to access low-cost data or free Wi-Fi to help facilitate increased access to all groups.

In relation to WP3 the focus will be on ensuring the analysis of the outcome measures can be compared across different demographic groups so that disparities can be identified. Exploring the factors that can influence differential uptake, use, and impact across the different clinics and demographic groups.

Outputs. An intervention which is accessible to people from a range of demographic backgrounds and that collects/analyses/acts on data suggesting differential uptake or impact of the digital health intervention by people from these demographic groups; and an understanding of the features of the

implementation model which promote participation by people from ethnic minority, disadvantaged, or older groups.